KA-TP-21-2025, P3H-25-051, IPPP/25/50, CERN-TH-2025-134

# One-Loop Calculations in Effective Field Theories with GoSam-3.0

Jens Braun,[1] Benjamin Campillo Aveleira,[1] Gudrun Heinrich,[1] Marius Höfer,[1]
Stephen P. Jones,[2] Matthias Kerner,[1] Jannis Lang,[1] Vitaly Magerya[3]

**1** Institute for Theoretical Physics, Karlsruhe Institute of Technology (KIT),
Wolfgang-Gaede-Str. 1, 76131 Karlsruhe, Germany
**2** Institute for Particle Physics Phenomenology, Durham University,
South Road, Durham DH1 3LE, U.K.
**3** Theoretical Physics Department, CERN, 1211 Geneva 23, Switzerland

jens.braun2@kit.edu, benjamin.campillo@kit.edu, gudrun.heinrich@kit.edu,
marius.hoefer@kit.edu, stephen.jones@durham.ac.uk, matthias.kerner@kit.edu,
jannis.lang@partner.kit.edu, vitaly.magerya@cern.ch

## Abstract

We present a major update of the one-loop generator GoSam, containing performance improvements as well as new features, in particular functionalities that facilitate calculations beyond the Standard Model in Effective Field Theory frameworks.

# 1   Introduction

Precision calculations are indispensable in the current particle physics landscape. Nowadays, the term "precision calculations" usually is associated with quantum corrections beyond the next-to-leading order (NLO) in the strong coupling, and the calculation of one-loop corrections is considered as a solved problem, since the basis integrals as well as automated IR subtraction procedures are known [1–3]. Automated tools for one-loop calculations in QCD or electroweak (EW) theory exist since quite some time [4–17] and partly have been refined to specialise on multi-particle final states or mixed QCD–EW corrections, see e.g. [18–20]. The automation of NLO corrections within theories beyond the Standard Model (SM) is also in a quite advanced phase. Model files can be generated through FEYNRULES [21, 22] or SMEFTFR [23–25] and imported through the Universal Feynman Output (UFO) format [26, 27], counterterms can be generated with the package NLOCT [28].

More recently, the automation of NLO QCD corrections within Standard Model Effective Field Theory (SMEFT) [29–33] has been presented in the framework of MG5_aMC@NLO [14] in Ref. [34], see also Ref. [35] for automation at leading order. While the MG5_aMC@NLO framework is fully automated, it can be useful to have additional tools at hand, not only for validation purposes. Additional functionalities, for example enhanced support for multi-leg processes or flexibility concerning the interface to NLO-capable Monte Carlo event generators such as HERWIG 7 [36, 37], HELAC-NLO [8], POWHEG [38–40], SHERPA [41, 42], or WHIZARD [43, 44]. The possibility to filter certain diagram classes or to control different EFT truncation schemes can also be useful for applications in beyond Standard Model (BSM) theories.

The upgrade to version 3 of the automated one-loop generator GOSAM [11, 12] offers such flexibility. With GOSAM-3.0 we present major improvements compared to previous versions: the installation process is much easier and the program is faster in both code generation and runtime. It also contains an upgrade of the stability test and rescue system for numerically problematic points. The functionalities to calculate QCD or electroweak corrections within extensions of the Standard Model, imported via UFO model files [26, 27], have also been extended. In particular, GOSAM-3.0 is capable of calculating QCD corrections within Effective Field Theories in a largely automated way, including different truncation options in the SMEFT expansion parameter. The code can also generate amplitudes involving 4-fermion operators; potential sign ambiguities have been carefully eliminated. In addition, the usage of Python filters to select classes of diagrams or couplings has been implemented in a user-friendly way. Therefore, the new version offers a lot of flexibility, which can be very useful for one-loop corrections in various types of BSM models, i.e. the new features are not limited to EFT applications.

This paper is structured as follows. In Section 2 we describe the installation and basic usage of GOSAM-3.0, Section 3 is dedicated to the description of the new features. In Section 4 we

describe some selected examples with the aim to demonstrate the new features, before we conclude in Section 5.

## 2   Installation and usage

### 2.1   Installation

GoSam-3.0 is available from the Git repository

    https://github.com/gudrunhe/gosam

or as a release tarball from

    https://github.com/gudrunhe/gosam/releases

The installation of GoSam-3.0 and its dependencies is handled by the Meson build system [45], which automatically downloads and compiles QGraf [46], Form [47,48], Ninja [49–51] and OneLOop [52]. This requires sufficiently modern Fortran, C and C++ compilers as well as Python ($\geq$ 3.9), Meson and (GNU) Make. To run the installation, the following commands should be executed in the cloned git repository or the extracted tarball:

```
$ meson setup build --prefix <prefix>
$ meson install -C build
```

This will install the main entry point, gosam.py, to <prefix>/bin, all dependencies are installed to subfolders like <prefix>/lib/GoSam/ to avoid conflicts with existing installations of any dependency. Executing gosam.py is sufficient for GoSam-3.0 to find all libraries and dependencies, no further setup of the environment is necessary.

By default only Ninja is included as reduction library, but Golem95 [53–55] is still available as an optional component. To install Golem95 as second reduction library, the argument –Dgolem95=true can be added to the setup command.

### 2.2   Usage

The generation of the matrix elements for a given process can be divided into three steps: diagram generation, code generation and compilation of the code. GoSam-3.0 needs an input file ("process card") in plain ASCII format where the process and the orders in a coupling constant to be calculated are defined. There are no restrictions on possible file names or extensions. For historical reasons GoSam process cards often use the file extensions *.in or *.rc. For example, a minimal process card for the process $e^+e^- \to t\bar{t}$ at NLO in QCD can look like this:

```
process_name=eett
process_path=eett_virtual
in=11,-11
out=6,-6
order=gs,0,2
```

It is also possible to generate and modify a template file for the process card instead of starting from scratch. This can be done by invoking the shell command

```
$ gosam.py --template <process_card>
```

The process card <process_card> generated in this way contains a description of all the available options. The options are also described in detail in Appendix E of the reference

manual coming with the code. It can be found in the doc subfolder of the code and is called refman.

The command

```
$ gosam.py <process_card>
```

will setup the directory structure under the path specified by process_path in the process card and generate the corresponding diagrams. After gosam.py has terminated, the build system has to be initialized. This is done by changing into the newly created process directory and executing the command

```
$ meson setup build --prefix <prefix> [-Doption=value]
```

where <prefix> is the location the process library is installed to. If no prefix is given meson will try to install the libraries in /usr/local. The option -Ddoc=true will generate the file doc/process.pdf containing the generated diagrams and other information about the process. The option -Dtest_executables=true produces a test program to evaluate the generated amplitude at a randomly generated phase space point.

After the process directory is configured, the generation and compilation of the Fortran 95 source code as well as the installation of the built process library can be triggered by executing

```
$ meson install -C <path/to/build>
```

where <path/to/build> is the relative path pointing to the directory build created in the previous step. The -C flag can be omitted when the command is started from inside build. By default, meson will fully utilize the CPU for compilation. If this is undesired, instead of invoking meson install directly one can instead use

```
$ meson compile [-j <jobs>] -C <path/to/build>
```

With the -j option, the number of jobs meson will run in parallel can be set. This command will only compile the source code, so in order to install the libraries one still has to call

```
$ meson install -C <path/to/build>
```

after the compilation is completed.

If -Dtest_executables=true has been set during setup, a test executable sampling the matrix element for a single random phase space point is now available in the subdirectory test.

For more details we refer to the reference manual in the doc folder of the code and the examples in the folder examples.

## 3 New features

### 3.1 Performance improvements

Version 3 of GoSam comes with many modernisations and performance improvements ranging from the installation of the GoSam code itself to the performance of the generated process libraries.

### 3.1.1 Compilation performance

In version 3 of GoSam, all code compilation and library building is handled by the modern, performance-oriented build system Meson. This enables GoSam to natively use all available

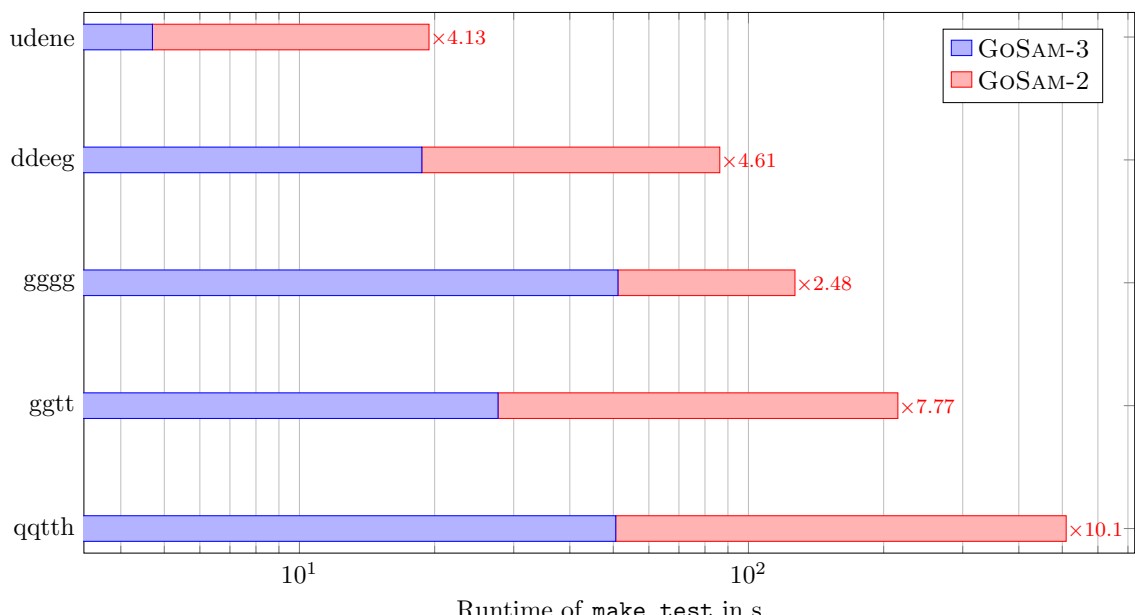

Figure 1: Runtime of `make test` for selected examples included in the GoSam distribution on an Intel Core i7-10700 8-core processor. The runtime of GoSam-2 relative to GoSam-3 is indicated next to each bar.

CPU cores on the host machine for reduction and compilation, significantly reducing all compile times on modern multicore systems. For the installation procedure of GoSam itself, this results in a reduction of the runtime by over 75% on an Intel Core i7-10700 8-core processor.

Similar improvements are also observable in the runtime of GoSam when generating a process library, which is now also handled largely by Meson. This allows the code generation of all helicities to be performed in parallel, as well as the compilation of all source files. The largest improvements are therefore visible in processes with many helicities on systems with many CPU cores. Figure 1 shows the runtime of some selected examples included in the GoSam distribution, which is dominated by the generation and compilation of the process library. These examples are representative of all the included examples, which overall show a substantial improvement in build time.

### 3.1.2  Runtime performance

Several improvements have also been implemented in the source code of the generated process libraries, significantly reducing evaluation times in many cases. This is achieved by optimising the calculation of the fundamental spinor brackets, which are now less frequently recomputed between helicities. Additionally, only the fundamental spinor brackets actually appearing in the amplitude are calculated. With these changes, the time to evaluate the amplitude is significantly reduced compared to previous versions of GoSam, while remaining numerically identical. This can be seen in Figure 2, which shows the evaluation time of the amplitude for some selected processes. Since the improvements are in the calculation of the fundamental spinor brackets and not in the amplitude itself, the performance gain is strongly dependent on the complexity of the amplitude. If a small part of the total time is spent on the fundamental spinor brackets, the improvement is only very mild. One example for this is `ggtt`, which only improved by 6%. For amplitudes where the speed improvements are more significant, a large portion of the time was previously spent in calculating the fundamental spinor brackets, an example for this is `tttt`. Here, the new version of GoSam evaluates the squared amplitude

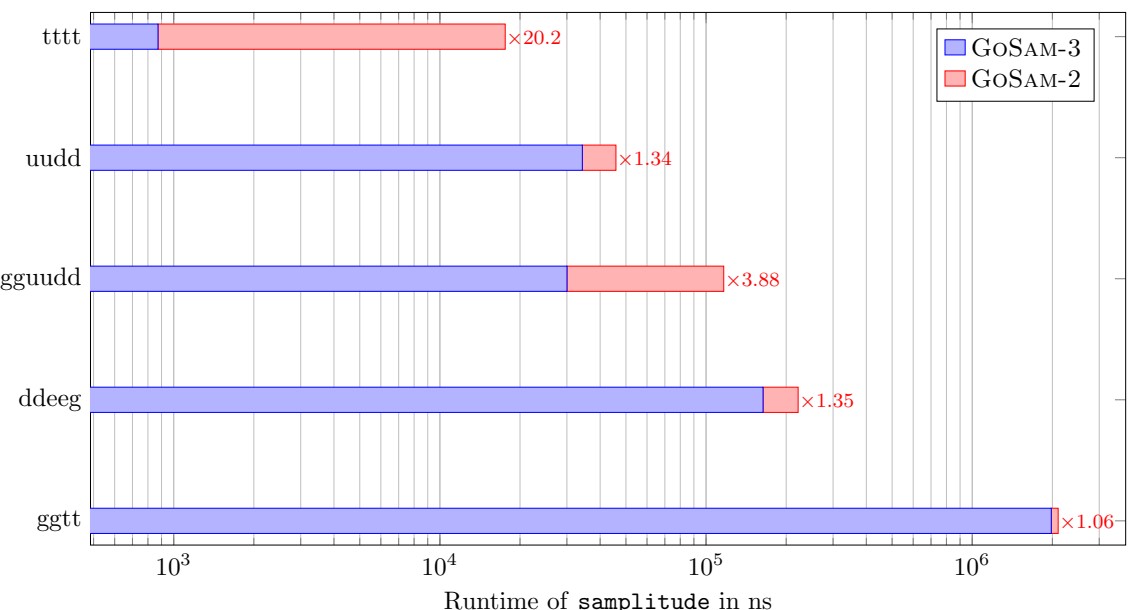

Figure 2: Runtime of a single evaluation of the squared amplitude for selected examples included in the GOSAM distribution on an Intel Core i7-10700 8-core processor. The runtime of GOSAM-2 relative to GOSAM-3 is indicated next to each bar.

more than 20 times faster compared to the previous version.

## 3.2   Rescue system

The rescue system of GOSAM has been extended to (optionally) utilise quadruple precision to re-evaluate unstable points. The stability of a point is assessed using three tests, known as the *pole test*, the *K-factor test* and the *rotation test*, respectively.

The pole test compares the general IR prediction for the single pole of a NLO QCD amplitude, $\mathcal{S}_{\text{IR}}$, with the singularity computed directly by GOSAM, $\mathcal{S}$,

$$\delta_{\text{pole}} = \left| \frac{\mathcal{S}_{\text{IR}} - \mathcal{S}}{\mathcal{S}_{\text{IR}}} \right|. \tag{1}$$

The estimate of the number of correct digits in the result is given by $P_{\text{pole}} = -\log_{10}(\delta_{\text{pole}})$. This stability check requires very little additional computational time as the matrix element does not need to be recomputed. The pole coefficient $\mathcal{S}_{\text{IR}}$ is calculated based on the assumption that the amplitude factorises into the product of the Born amplitude times the IR insertion operator [2,56] in the IR singular limit. Therefore the pole check also works for BSM models as long as this property still holds. A similar check can be used for the Born result of loop-induced processes, where the single pole is expected to vanish. In the loop-induced case, we define

$$\delta_{\text{pole}} = \left| \frac{\mathcal{S}}{\mathcal{A}} \right|. \tag{2}$$

where $\mathcal{A}$ is the finite part of the amplitude.

The *K-factor test* computes the ratio of the finite part of the NLO amplitude, $\mathcal{A}$, and the corresponding Born amplitude, $\mathcal{B}$,

$$K = \left| \frac{\mathcal{A}}{\mathcal{B}} \right|. \tag{3}$$

For loop-induced processes, GoSam computes only the Born amplitude, we define the $K$-factor as the absolute magnitude of a dimensionless quantity consisting of the finite part of the amplitude multiplied by a power of the largest Mandelstam invariant present, $s_{\text{max}}$,

$$K = |\mathcal{A} \cdot s_{\text{max}}^{N-4}|, \tag{4}$$

where $N$ is the number of external legs in the process.

The rotation test [50] exploits the invariance of scattering amplitudes under an azimuthal rotation about the beam axis. The finite part of the amplitude, $\mathcal{A}$, is compared to the value of the amplitude obtained after rotating the input kinematics in the azimuthal plane, $\mathcal{A}_{\text{rot}}$,

$$\delta_{\text{rot}} = 2 \left| \frac{\mathcal{A}_{\text{rot}} - \mathcal{A}}{\mathcal{A}_{\text{rot}} + \mathcal{A}} \right|. \tag{5}$$

The estimate of the number of correct digits in the result is given by $P_{\text{rot}} = -\log_{10}(\delta_{\text{rot}})$. We also define a dimensionless quantity capturing the absolute difference between the amplitude computed before and after rotation,

$$\Delta_{\text{rot}} = |(\mathcal{A}_{\text{rot}} - \mathcal{A}) \cdot s_{\text{max}}^{N-4}| \tag{6}$$

The default procedure to assess the stability of a point uses the pole and $K$-factor checks followed by a rotation check if necessary. Firstly, $P_{\text{pole}}$ and $K$ are computed,

1. $P_{\text{pole}} < $ `PSP_chk_th2` $\rightarrow$ rescue point,

2. `PSP_chk_kfactor` $< K \rightarrow$ rotation test,

3. $P_{\text{pole}} < $ `PSP_chk_th1` $\rightarrow$ rotation test,

4. $\rightarrow$ accept point,

If a rotation test is required, $P_{\text{rot}}$ and $\Delta_{\text{rot}}$ are computed,

1. $P_{\text{rot}} < $ `PSP_chk_th3` $\rightarrow$ rescue point,

2. `PSP_chk_rotdiff` $< \Delta_{\text{rot}} \rightarrow$ rescue point,

3. $\rightarrow$ accept point.

By default, if the above checks trigger the rescue system then the point will be recomputed with the reduction library specified in `reduction_interoperation_rescue` if an alternative to the default reduction library is available (i.e. if GoSam was compiled with the Golem95 option enabled and the default is Ninja) and a pole check followed by a rotation check will be performed. If these checks fail or the rescue system is disabled, the point is discarded and a precision of $-10$ is returned.

If `extensions=quadruple` is set during the GoSam generation phase, then the rescue system will instead recompute the unstable point in quadruple precision and calculate

$$\delta_{\text{qd}} = 2 \left| \frac{\mathcal{A} - \mathcal{A}_{\text{q}}}{\mathcal{A} + \mathcal{A}_{\text{q}}} \right|, \qquad \delta_{\text{qdrot}} = 2 \left| \frac{\mathcal{A}_{\text{rot}} - \mathcal{A}_{\text{q}}}{\mathcal{A}_{\text{rot}} + \mathcal{A}_{\text{q}}} \right|, \tag{7}$$

$$P_{\text{qd}} = -\log_{10}(\delta_{\text{q}}), \qquad P_{\text{qdrot}} = -\log_{10}(\delta_{\text{qdrot}}), \tag{8}$$

where $\mathcal{A}_q$ is the finite part of the amplitude computed in quadruple precision. If both `PSP_chk_th4` $< P_{\text{qd}}$ and `PSP_chk_th4` $< P_{\text{qdrot}}$ the point is accepted. If this precision threshold is not met,

then the input kinematics are rotated in the azimuthal plane and the amplitude is recomputed in quadruple precision, $\mathcal{A}_{\text{qrot}}$. The quantities

$$\delta_{\text{qqrot}} = 2\left|\frac{\mathcal{A}_{\text{qrot}} - \mathcal{A}_{\text{q}}}{\mathcal{A}_{\text{qrot}} + \mathcal{A}_{\text{q}}}\right|, \qquad P_{\text{qqrot}} = -\log_{10}(\delta_{\text{qqrot}}), \qquad (9)$$

are evaluated. If `PSP_chk_th5` $< P_{\text{qqrot}}$ the point is accepted. All remaining points are discarded and a precision of $-10$ is returned.

For loop-induced processes, the stability and rescue procedure are applied to the Born result, the thresholds `PSP_chk_th*` are replaced by `PSP_chk_li*`, `PSP_chk_kfactor` is replaced by `PSP_chk_kfactor_li` and `PSP_chk_rotdiff` is replaced by `PSP_chk_rotdiff_li`.

Typically, the fraction of unstable points does not exceed the percent range and therefore using `extensions=quadruple` is the recommended rescue setup, as it is not too costly in runtime.

## 3.3 Calculations within SMEFT or HEFT

### 3.3.1 UFO models for EFT calculations

GoSam does not come with any built-in EFT models. For a calculation based on an EFT the user has to provide the model through the generic UFO interface, see section 3.10.1 of the reference manual. GoSam is able to handle $n$-point vertices, with $n > 4$, and 4-fermion interactions. Note that, when no additional order besides the usual QCD and QED orders is specified for the vertices, GoSam will treat all interactions equally, considering only their assigned power with respect to the perturbative expansion in the strong and electroweak/QED coupling. In most cases a distinction between SM and non-SM interactions is desirable, which in UFO syntax is conventionally handled through additional coupling orders. GoSam reserves two special order names, `NP` and `QL`. The former is used to assign an order to a coupling with respect to the power counting of the EFT, for example factors of $1/\Lambda$ in SMEFT. The latter can be used to assign a loop-order to the coupling, taking into account a potential loop-suppression of EFT operators, as explained in more detail in section 3.3.3 below.

A special remark has to be made about double, or in general multiple, insertions of EFT operators. Per default GoSam will generate diagrams with multiple insertions of non-SM vertices, if they are present in the model. However, in a SMEFT context this leads to inconsistencies when at the same time operators of even higher dimension are missing in the model. For example, a double insertion of dimension 6 operators is at the same order as a single insertion of a dimension 8 operator. To be fully consistent, both cases would have to be included. The user can avoid such problems by using the Python diagram filters to single out diagrams with at most one SMEFT vertex:

```
filter.lo=lambda d: d.order('NP')<=1
filter.nlo=lambda d: d.order('NP')<=1
filter.ct=lambda d: d.order('NP')<=1
```

In this example we assume that the leading EFT operators are flagged by `NP=1` in the UFO model.

### 3.3.2 Multi-fermion operators

The treatment of diagram signs has been extended to handle vertices containing more than two fermions. The determination of the diagram sign is based on the algorithm of Ref. [57], which requires tracing the fermion lines of a diagram. This tracing is unambiguous for vertices containing only two fermions, but requires additional information on how the legs of a vertex

are connected when more than two fermions are involved. GOSAM reads this information from the analytical vertex structure supplied by the UFO model. If the leg connections of a vertex are ambiguous, i.e. the analytical expression is a sum of two or more Lorentz structures connecting the legs differently, it is split into multiple vertices such that the leg connections are unambiguous for every vertex. After this procedure, GOSAM's reduction machinery is able to handle the resulting diagrams normally.

### 3.3.3  Truncation orders in SMEFT

SMEFT is an expansion in inverse powers of the scale of new physics $\Lambda$,

$$\mathcal{L} = \mathcal{L}_{\text{SM}} + \sum_{d>4} \sum_{i_d} \frac{C_{i_d}^d}{\Lambda^d} O_{i_d}^d \,, \tag{10}$$

where $d$ denotes the canonical dimension of the operators $O_{i_d}^d$, with corresponding Wilson coefficients $C_{i_d}^d$. In order to calculate physical quantities one has to truncate the SMEFT expansion at a specific order. Precisely how this truncation is defined is not free of ambiguities, since it can be implemented on the level of the amplitudes or at the level of squared matrix elements. For this reason GOSAM supports different truncation options for SMEFT calculations by setting `enable_truncation_orders=true` in the process card, provided the process setup and model used meet some requirements:

- The model is provided in the UFO format.

- All of the model's SMEFT operators are of the same dimension, with corresponding coupling order set to NP=1. Some SMEFT models might assign NP=2 to the dimension 6 terms, accounting for the fact that technically also two dimension 5 terms exist in the SMEFT, which are often dropped.[1] In this case the user should adjust the model accordingly.

- The GOSAM process card has to specify the property `order_names=NP`. Additional order names, like e.g. QCD or QED are optional.

In some cases the user might want to take into account an intrinsic loop suppression of certain operators. Couplings arising from those should be flagged by the additional order QL=1 in the UFO model. We can now decompose any amplitude in the following way:

$$\mathcal{M}^\ell = \underbrace{\mathcal{M}_{\text{SM}}^\ell}_{\text{NP=0}} + \underbrace{\overbrace{\frac{\mathcal{M}_6^\ell}{\Lambda^2}}^{\text{QL=0}} + \overbrace{\frac{\bar{\mathcal{M}}_6^\ell}{\Lambda^2}}^{\text{QL=1}}}_{\text{NP=1}} \,, \tag{11}$$

where $\ell = 0, 1$ denotes the type of diagram topology, tree or 1-loop. $\mathcal{M}_6^\ell$ is the contribution of diagrams with a single insertion of a dim-6 operator which is not loop-suppressed and $\bar{\mathcal{M}}_6^\ell$ contains those which are loop-suppressed.[2] Subsequently, there are different ways of treating the truncation of the amplitude to calculate physical quantities based on squared matrix elements. Currently nine different truncation options are implemented in GOSAM, which will be explained in detail below. They can be chosen at runtime by means of the variable `EFTcount`. Possible values are shown in Table 1.

---

[1]There are only two lepton-number-violating operators. Experimental findings suggest them to be extremely suppressed.

[2]GOSAM does not make any assumptions about implicit factors (e.g. couplings and/or factors of $\pi$) contained in the Wilson coefficient of loop-suppressed operators. The Wilson coefficients are taken exactly as they are defined in the UFO model and no additional loop-suppression factor is added to the resulting amplitudes by GOSAM.

| EFTcount | loop-suppression | truncation | |
|:---:|:---:|:---|:---|
| 0 | — | $\text{SM}^2$ | pure SM (default setting) |
| 1 | no | $\text{SM}^2 + \text{SM} \otimes \text{dim-6}$ | linear truncation |
| 2 | no | $(\text{SM} + \text{dim-6})^2$ | quadratic truncation |
| 3 | no | $\text{SM} \otimes \text{dim-6}$ | linear coefficient |
| 4 | no | $\text{dim-6}^2$ | quadratic coefficient |
| 11 | yes | $\text{SM}^2 + \text{SM} \otimes \text{dim-6}$ | linear truncation |
| 12 | yes | $(\text{SM} + \text{dim-6})^2$ | quadratic truncation |
| 13 | yes | $\text{SM} \otimes \text{dim-6}$ | linear coefficient |
| 14 | yes | $\text{dim-6}^2$ | quadratic coefficient |

Table 1: Possible choices for the variable `EFTcount` and corresponding truncation. $A \otimes B \equiv 2 \, \text{Re} \left\{ A^\dagger B \right\}$.

In the following we will show the structure of the results returned by GoSam for the Born matrix element and the virtual corrections. We use the notation $A \otimes B \equiv 2 \, \text{Re} \left\{ A^\dagger B \right\}$ and drop the $\Lambda^{-2}$ for reasons of legibility. Since loop-induced processes require a slightly different treatment they are discussed in section 3.3.4 below.

`EFTcount=0`: **$\text{SM}^2$**    This option discards any higher dimensional operators and returns just the SM result. This is the default.

$$\text{Born:} \qquad \left| \mathcal{M}_{\text{SM}}^0 \right|^2, \tag{12}$$

$$\text{Virtual:} \qquad \mathcal{M}_{\text{SM}}^0 \otimes \mathcal{M}_{\text{SM}}^1. \tag{13}$$

`EFTcount=1`: **$\text{SM}^2$ + SM × dim-6, ignoring loop-suppression**    All SMEFT operators are treated equally and no kind of loop-suppression is assumed for any of them. $\mathcal{M}_6$ and $\bar{\mathcal{M}}_6$ thus enter the expressions for the squared matrix elements in exactly the same way. We have

$$\text{Born:} \qquad \left| \mathcal{M}_{\text{SM}}^0 \right|^2 + \mathcal{M}_{\text{SM}}^0 \otimes \left( \mathcal{M}_6^0 + \bar{\mathcal{M}}_6^0 \right), \tag{14}$$

$$\text{Virtual:} \qquad \mathcal{M}_{\text{SM}}^0 \otimes \mathcal{M}_{\text{SM}}^1 + \mathcal{M}_{\text{SM}}^0 \otimes \left( \mathcal{M}_6^1 + \bar{\mathcal{M}}_6^1 \right) + \left( \mathcal{M}_6^0 + \bar{\mathcal{M}}_6^0 \right) \otimes \mathcal{M}_{\text{SM}}^1. \tag{15}$$

`EFTcount=2`: **$(\text{SM} + \text{dim-6})^2$, ignoring loop-suppression**    This option essentially is "no truncation" in the sense that the full available amplitude is simply squared.

$$\text{Born:} \qquad \left| \mathcal{M}_{\text{SM}}^0 + \mathcal{M}_6^0 + \bar{\mathcal{M}}_6^0 \right|^2, \tag{16}$$

$$\text{Virtual:} \qquad \left( \mathcal{M}_{\text{SM}}^0 + \mathcal{M}_6^0 + \bar{\mathcal{M}}_6^0 \right) \otimes \left( \mathcal{M}_{\text{SM}}^1 + \mathcal{M}_6^1 + \bar{\mathcal{M}}_6^1 \right). \tag{17}$$

`EFTcount=3`: **SM×dim-6, ignoring loop-suppression**    This is the linear dim-6 contribution, i.e. the part of the squared matrix element which is $\mathcal{O}(\Lambda^{-2})$.

$$\text{Born:} \qquad \mathcal{M}_{\text{SM}}^0 \otimes \left( \mathcal{M}_6^0 + \bar{\mathcal{M}}_6^0 \right), \tag{18}$$

$$\text{Virtual:} \qquad \mathcal{M}_{\text{SM}}^0 \otimes \left( \mathcal{M}_6^1 + \bar{\mathcal{M}}_6^1 \right) + \left( \mathcal{M}_6^0 + \bar{\mathcal{M}}_6^0 \right) \otimes \mathcal{M}_{\text{SM}}^1. \tag{19}$$

`EFTcount=4`: **(dim-6)$^2$, ignoring loop-suppression**    The dim-6 part of the amplitude squared:

$$\text{Born:} \qquad \left|\mathcal{M}_6^0 + \bar{\mathcal{M}}_6^0\right|^2, \tag{20}$$

$$\text{Virtual:} \qquad \left(\mathcal{M}_6^0 + \bar{\mathcal{M}}_6^0\right) \otimes \left(\mathcal{M}_6^1 + \bar{\mathcal{M}}_6^1\right). \tag{21}$$

`EFTcount=11`: **SM$^2$ + SM × dim-6, with loop-suppression**    "With loop-suppression" means that the operators which are assumed to be loop-generated in a UV-complete model are treated as introducing an additional loop order to the diagrams they are contributing to. Effectively, this results in $\bar{\mathcal{M}}_6^0$ being considered a 1-loop contribution at the same (loop and NP) order as $\mathcal{M}_6^1$. $\bar{\mathcal{M}}_6^1$ then corresponds to 2-loops and is consequently dropped.

$$\text{Born:} \qquad \left|\mathcal{M}_{\text{SM}}^0\right|^2 + \mathcal{M}_{\text{SM}}^0 \otimes \mathcal{M}_6^0, \tag{22}$$

$$\text{Virtual:} \qquad \mathcal{M}_{\text{SM}}^0 \otimes \mathcal{M}_{\text{SM}}^1 + \mathcal{M}_{\text{SM}}^0 \otimes \mathcal{M}_6^1 + \mathcal{M}_6^0 \otimes \mathcal{M}_{\text{SM}}^1 + \left[\mathcal{M}_{\text{SM}}^0 \otimes \bar{\mathcal{M}}_6^0\right]. \tag{23}$$

The term in square brackets is then a tree-structure (0-loop topologies) contributing to the 1-loop order.

`EFTcount=12`: **(SM + dim-6)$^2$, with loop-suppression**    Due to consideration of the loop-suppression this option is not a simple square anymore.

$$\text{Born:} \qquad \left|\mathcal{M}_{\text{SM}}^0 + \mathcal{M}_6^0\right|^2, \tag{24}$$

$$\text{Virtual:} \qquad \left(\mathcal{M}_{\text{SM}}^0 + \mathcal{M}_6^0\right) \otimes \left(\mathcal{M}_{\text{SM}}^1 + \mathcal{M}_6^1\right) + \left[\left(\mathcal{M}_{\text{SM}}^0 + \mathcal{M}_6^0\right) \otimes \bar{\mathcal{M}}_6^0\right]. \tag{25}$$

There is no term $\left|\bar{\mathcal{M}}_6^0\right|^2$ in the virtual part, as this would be a 2-loop structure, despite being constructed solely from diagrams with tree topology.

`EFTcount=13`: **SM × dim-6, with loop-suppression**    The linear dim-6 contribution, but treating $\bar{\mathcal{M}}_6^0$ as a 1-loop order object.

$$\text{Born:} \qquad \mathcal{M}_{\text{SM}}^0 \otimes \mathcal{M}_6^0, \tag{26}$$

$$\text{Virtual:} \qquad \mathcal{M}_{\text{SM}}^0 \otimes \mathcal{M}_6^1 + \mathcal{M}_6^0 \otimes \mathcal{M}_{\text{SM}}^1 + \left[\mathcal{M}_{\text{SM}}^0 \otimes \bar{\mathcal{M}}_6^0\right]. \tag{27}$$

`EFTcount=14`: **(dim-6)$^2$, with loop-suppression**    The squared dim-6 part of the amplitude, considering the extra loop order of $\bar{\mathcal{M}}_6^0$ and $\bar{\mathcal{M}}_6^1$:

$$\text{Born:} \qquad \left|\mathcal{M}_6^0\right|^2, \tag{28}$$

$$\text{Virtual:} \qquad \mathcal{M}_6^0 \otimes \mathcal{M}_6^1 + \left[\mathcal{M}_6^0 \otimes \bar{\mathcal{M}}_6^0\right]. \tag{29}$$

Note that, when the model does not contain any loop-suppressed operators, we have $\bar{\mathcal{M}}_6^\ell \equiv 0$ and the truncation options 11, 12, 13, 14 return the same results as options 1, 2, 3, 4, respectively.

### 3.3.4  Loop-induced processes

Processes which are loop induced in the SM require a special treatment. In some cases the inclusion of EFT operators generates tree-level contributions to such processes. A famous example is the decay of the Higgs boson into two gluons, which in the SM is mediated via a top-quark loop. When adding the Higgs-gluon operator $\mathcal{O}_{\phi G} = \phi^\dagger \phi\, G_{\mu\nu}^a G^{a,\mu\nu}$ to the theory the decay can be generated at tree-level.

In order to consistently define the process one has to distinguish two scenarios:

1. Tree-level contributions are generated by tree-level EFT operators, that is operators which are not considered loop-suppressed.

2. Tree-level contributions are generated by loop-suppressed EFT operators only.

In the first scenario the process is not actually loop-induced, and the process can be set up in the usual way with a tree-level Born. A requirement is that the QCD and/or QED orders of the EFT operators are consistent with the `order` statement in the process configuration file. In this case all truncation orders can be defined as above, with $\mathcal{M}^0_{SM} \equiv 0$. Note, however, that this means that the leading contributions to the process are tree times 1-loop interferences at dim-6 and squared tree-level contributions at dim-$6^2$. The actual SM part is then only subleading in perturbation theory at 1-loop squared and will not even be calculated by GoSam.

In that sense the second scenario is more interesting. In this case the process is treated as loop-induced and the loop-suppressed EFT diagrams with tree-level topology are considered as of the same level as the 1-loop (SM-)contributions. Since GoSam cannot know a priori if such EFT diagrams exist the user has to explicitly set the flag

    loop_suppressed_Born=true

in the process card. As a consequence only the `EFTcount` options 0 and 11 to 14, that is the ones considering loop-suppression, are defined for loop-induced processes. The tree and 1-loop contributions to the process can then be written as (dropping the $\Lambda^{-2}$ as above)

$$\mathcal{M}^0 = \bar{\mathcal{M}}^0_6, \qquad\qquad \mathcal{M}^1 = \mathcal{M}^1_{SM} + \mathcal{M}^1_6, \qquad (30)$$

respectively. The tree-level contains diagrams with loop-suppressed operators, only, while the 1-loop level comprises SM loop diagrams and loop diagrams with single insertions of tree-type EFT operators. There are no loop-diagrams with loop-suppressed operators, as they are discarded as subleading. The structure of the results for the loop-induced Born returned by GoSam is summarized in the following. In all cases the loop-suppressed $\bar{\mathcal{M}}^0_6$ is being considered a 1-loop contribution at the same loop order as $\mathcal{M}^1_{SM}$ and $\mathcal{M}^1_6$.

`EFTcount=0`: **SM$^2$**   This option discards any higher dimensional operator and returns just the SM result. This is the default.

Loop-ind. Born:     $\left| \mathcal{M}^1_{SM} \right|^2.$ $\qquad\qquad\qquad\qquad\qquad\qquad\qquad$ (31)

`EFTcount=11`: **SM$^2$ + SM × dim-6, with loop-suppression**   Truncation at linear order in $\Lambda^{-2}$.

Loop-ind. Born:     $\left| \mathcal{M}^1_{SM} \right|^2 + \mathcal{M}^1_{SM} \otimes \left( \mathcal{M}^1_6 + \bar{\mathcal{M}}^0_6 \right).$ $\qquad\qquad$ (32)

`EFTcount=12`: **(SM + dim-6)$^2$, with loop-suppression**   The truncation option including dim-$6^2$ terms.

Loop-ind. Born:     $\left| \mathcal{M}^1_{SM} + \mathcal{M}^1_6 + \bar{\mathcal{M}}^0_6 \right|^2.$ $\qquad\qquad\qquad\qquad$ (33)

`EFTcount=13`: **SM × dim-6, with loop-suppression**   The linear dim-6 contribution.

Loop-ind. Born:     $\mathcal{M}^1_{SM} \otimes \left( \mathcal{M}^1_6 + \bar{\mathcal{M}}^0_6 \right).$ $\qquad\qquad\qquad\qquad$ (34)

**EFTcount=14: (dim-6)$^2$, with loop-suppression**  The squared dim-6 contribution:

$$\text{Loop-ind. Born:} \qquad \left| \mathcal{M}_6^1 + \bar{\mathcal{M}}_6^0 \right|^2 . \tag{35}$$

Note that above options are still well defined when the model does not contain any loop-suppressed operators. We then have $\bar{\mathcal{M}}_6^0 \equiv 0$ and only genuine 1-loop squared topologies appear.

### 3.3.5  Calculations in HEFT

Higgs Effective Field Theory (HEFT) [58–66] organises the power counting in terms of the chiral dimension instead of the canonical dimension of operators, as SMEFT does. As the chiral dimension is directly related to the loop order of an operator, the loop-suppression mentioned above is an integral feature of HEFT. Operators from the leading HEFT Lagrangian $\mathcal{L}_2$ are considered "tree-level operators", those from the next-to-leading (in the power counting) Lagrangian $\mathcal{L}_4$ "one-loop operators" and so on.[3] The full SM is a subset of $\mathcal{L}_2$.

Each insertion of a vertex coming from $\mathcal{L}_{2n}$ raises the loop order of a given diagram by $n-1$. For example, a single $\mathcal{L}_4$ vertex in a diagram with tree topology will make it a one-loop order diagram, contributing at the same level as a genuine one-loop diagram with only vertices from $\mathcal{L}_2$.

GOSAM is able to perform calculations to NLO in HEFT, when in the corresponding UFO model all vertices from $\mathcal{L}_2$ are tagged with (NP=0, QL=0), vertices from $\mathcal{L}_4$ with (NP=1, QL=1). Vertices from $\mathcal{L}_6$ are not considered, as they are of two-loop order. The amplitude is then assembled consistently when setting EFTcount=12. In the above notation, $\mathcal{M}_{\text{SM}}^\ell$ then corresponds to *all* contributions from $\mathcal{L}_2$, including anomalous ones, and $\bar{\mathcal{M}}_6^0$ to contributions with tree topology and a single insertion of a $\mathcal{L}_4$ vertex. $\mathcal{M}_6^\ell$ is not present in the HEFT setup.

### 3.3.6  Renormalisation of Wilson coefficients

GOSAM is able to provide renormalised amplitudes at NLO QCD in the SM. See [11] for details about the construction of the corresponding counterterms. Renormalisation in an EFT context is a non-trivial task and not fully automatized in GOSAM. In general GOSAM therefore provides unrenormalised amplitudes when considering an EFT. However, under certain circumstances GOSAM is able to calculate the required counterterms for the 1-loop QCD renormalisation, just as in the pure SM. The necessary condition for this to work is that there are no contributions of the EFT operators to the renormalisation of SM parameters and fields. In other words, all additional UV divergences at $\mathcal{O}\left(\Lambda^{-2}\right)$ can be absorbed by renormalising the Wilson coefficients of the EFT operators alone, without the need to change the counterterms of SM objects. Internally GOSAM uses its infrastructure for the generation of the Born amplitude, by replacing occurrences of Wilson coefficients within each diagram by their corresponding counterterm, $C_i \rightarrow \delta C_i$. The result is expanded in a way that ensures that each contribution to the counterterm amplitude only has a single insertion of a counterterm.

The counterterms related to the Wilson coefficients have to be provided by the user. This can be done in a convenient way by means of the UFO interface, as explained in section 5.4 of the UFO2.0 manual [27]. Analogously to an ordinary Vertex object a counterterm vertex CTVertex can be defined, which exactly originates from the replacement $C_i \rightarrow \delta C_i$ mentioned above. As an example consider a simplified version of SMEFT with just the two operators

$$O_{\phi G} = \left( \phi^\dagger \phi \right) G_{\mu\nu}^a G^{a,\mu\nu} , \qquad\qquad O_{t\phi} = \left( \phi^\dagger \phi \right) \bar{Q}_L t_R \tilde{\phi} . \tag{36}$$

---

[3]The subscript indicates the chiral dimension. It is always an even number. For the technical details of HEFT we refer to the given references.

The latter generates (among others) an anomalous $t\bar{t}H$ vertex, which in the UFO can be defined as

```
V_1 = Vertex(
    name = 'V_1',
    particles = [ P.t__tilde__, P.t, P.H ],
    color = [ 'Identity(1,2)' ],
    lorentz = [ L.FFS1, L.FFS2 ],
    couplings = {(0,0):C.GC_1, (0,1):C.GC_2})
```

The vertex has a single colour structure, the identity, and two separate Lorentz structures and couplings, defined in the UFO's `lorentz.py` and `couplings.py`, respectively.[4] The $\overline{\text{MS}}$ counterterm for the Wilson coefficient $C_{t\phi}$ is given by

$$
\delta C_{t\phi} = \frac{\alpha_s}{2\pi} \frac{(4\pi)^\epsilon}{\Gamma(1-\epsilon)} \left(\frac{\mu_R^2}{\mu_{\text{EFT}}^2}\right)^\epsilon \left(\frac{1}{\epsilon}\left(-2C_{t\phi} + 8y_t C_{\phi G}\right)\right.
$$
$$
\left. + 1_{\text{DRED}}\left(-\frac{2}{3}C_{t\phi} + \frac{8}{3}y_t C_{\phi G}\right)\right) + \mathcal{O}(\epsilon), \quad (37)
$$

where $1_{\text{DRED}} = 1$, if the calculation is performed in DRED (GoSam's default) and $1_{\text{DRED}} = 0$ in the 't Hooft–Veltman scheme. $\mu_{\text{EFT}}$ is the scale at which the Wilson coefficients are renormalised, which can in general be different from the renormalisation scale $\mu_R$ of the strong coupling. The corresponding counterterm vertex is defined in `CT_vertices.py`:

```
CTV_1 = CTVertex(
    name = 'CTV_1',
    type = 'UV',
    particles = [ P.t__tilde__, P.t, P.H ],
    color = [ 'Identity(1,2)' ],
    lorentz = [ L.FFS3, L.FFS4 ],
    loop_particles = [ [ [ P.g ], [P.t] ] ],
    couplings = {(0,0,0):C.UVGC_1, (0,1,0):C.UVGC_2})
```

GoSam only makes use of counterterms of the type `'UV'`, while the UFO model in general can also provide $R2$ counterterms. However, those are not needed by GoSam. The counterterm vertex has the same colour and Lorentz structure as the ordinary vertex. The additional list `loop_particles` contains information about the type of particles appearing in the loops related to the derivation of the counterterm, with the intention to give the user an extra way to filter counterterm vertices. Currently GoSam does not make use of this feature, so this list can be ignored and treated as a dummy object. The two couplings are given by

```
UVGC_1 = Coupling(
    name = 'UVGC_1',
    value = '(complex(0,1)*Lam*(Ctphi_CT))/(2.*Gf)',
    order = {'NP':1, 'QED':1, 'QCD':2})
```

and

```
UVGC_2 = Coupling(
    name = 'UVGC_2',
    value = '(Ctphi_CT*complex(0,1)*Lam)/(2.*Gf)',
    order = {'NP':1, 'QED':1, 'QCD':2})
```

---

[4]See the UFO2.0 manual [27] for a detailed explanation of the syntax.

defined in `CT_couplings.py`. Here `Lam` $= \Lambda^{-2}$ is the SMEFT NP scale, and $G_F^{-1} \propto v^2$, with $v$ the Higgs vacuum expectation value, comes from the operator definition. The special `CTParameter` object `Ctphi_CT` is defined in `CT_parameters.py`:

```
Ctphi_CT = CTParameter(
   name = 'Ctphi_CT',
   type = 'real',
   value = {
      -1:'aS/2/cmath.pi*(-2*Ctphi+8*yt*CphiG)',
      0:'dred*aS/2/cmath.pi*(-2/3*Ctphi+8/3*yt*CphiG)'
   })
```

It reflects the structure of $\delta C_{t\phi}$ given in (37). Note that the UFO format permits to omit the definition of a `CTParameter` object by directly defining the `value` of the coupling as a Python dict instead. Two comments are in order: First, GOSAM assumes the strong coupling factor $\alpha_s/2\pi$ to be explicitly contained in the definition of the counterterm. Secondly, GOSAM assumes the counterterms of the Wilson coefficients to be in the $\overline{\text{MS}}$ scheme, with the scale factor $\left(\mu_R^2/\mu_{\text{EFT}}^2\right)^\epsilon$. GOSAM automatically expands this factor to obtain the appropriate logarithmic terms, which therefore do not have to be defined explicitly. The only requirement is the presence of a parameter `mueft` (corresponding to $\mu_{\text{EFT}}$) in the UFO's `parameters.py` file.

## 3.4   Updates in the BLHA interface

Already since GOSAM-1.0 the Binoth-Les-Houches-Accord (BLHA) standard for interfacing GOSAM with Monte Carlo (MC) event generators is supported. In the following, we will call this use-case of GOSAM the "OLP-mode". Both the BLHA1 [67] and the BLHA2 [68] standard are available. Note that with BLHA2 it is possible to not only pass one-loop amplitudes to the MC program, but also tree-level amplitudes for the Born and real-radiation processes, as well as spin- and colour-correlated tree amplitudes required by IR-subtraction schemes. GOSAM is therefore able to provide all necessary matrix elements for a full NLO QCD calculation. This is particularly useful for calculations within an EFT such as SMEFT or HEFT, or in general in BSM models, as any kind of selection or filtering on the contributing diagrams can be done consistently within GOSAM, for all components (Born, real and virtual corrections) of the calculation.

The main updates to the BLHA interface in GOSAM-3.0 concern improvements in stability of the tree-level amplitudes when used as the real-radiation matrix elements by the MC, the implementation of the SMEFT truncation features also in OLP-mode, and additions required to use GOSAM in conjunction with WHIZARD [43, 44]. The revised version of the interface has been applied successfully in the calculation of the NLO QCD corrections to double Higgs production in VBF with WHIZARD in [69].

We refer to chapter 9 of the reference manual for a detailed description of the BLHA interface implemented in GOSAM.

## 4   Examples

In this section we describe the examples added in version 3.0 in order to demonstrate some of the new features. They can be found in the folder `examples` of the GOSAM-3.0 release. The other examples are described in detail in Refs. [11, 12].

## 4.1 VBF production of a Higgs boson in unitary gauge

The example udhud_unitary demonstrates how to calculate the process $u\bar{d} \rightarrow Hu\bar{d}$, a sub-process from Higgs production in vector boson fusion (VBF) both in unitary gauge and in Feynman gauge, at NLO QCD, based on the corresponding UFO model files. This example demonstrates GOSAM's ability to perform calculations in unitary gauge. Note that GOSAM cannot handle loop-integrals in which the power of the loop-momentum in the numerator exceeds the number of propagators in the loop by more than one. This can easily happen when gauge propagators are treated in unitary gauge. However, for the example at hand this is not the case and the calculation can be performed.

## 4.2 $e^+e^- \rightarrow ZH$ in SMEFT

The example eeZH_SMEFT calculates the process $e^+e^- \rightarrow ZH$ at leading order, including the dimension-6 SMEFT operators $O_{\phi B}$, $O_{\phi W}$, $O_{\phi WB}$. It uses a modified version of the UFO model SMEFTsim_U35_MwScheme_UFO [35,70], with only those three operators included. The purpose of the example is to showcase how to use GOSAM and its UFO interface to perform calculations in SMEFT, focusing in particular on the different truncation options explained in section 3.3.3 (excluding the ones with loop-counting).

## 4.3 $H \rightarrow b\bar{b}$ in SMEFT

The example Hbb_SMEFT describes Higgs boson decay into a massive $b$-quark pair, including a Yukawa-type operator $O_{b\phi}$ and the Higgs-gluon operator $O_{\phi G}$. The $b$-quarks are renormalised in the on-shell scheme. Similarly to the previous example the use of different truncation options is shown, but the main purpose is to demonstrate how to provide counterterms for the Wilson coefficients calculated by hand to GOSAM through the UFO interface (see section 3.3.6). The underlying UFO model has been created using the SMEFTFR Mathematica package [23–25]. This package does not implement $O_{b\phi}$ directly, but rather a more general version with a Wilson coefficient representing a three-by-three matrix in flavour space. In order to reduce the complexity of the model all entries but the $(3,3)$ component have been removed, as have the vertices coupling the Higgs to the first two generations of quarks and leptons. In addition some redundant parameters have been removed and the sign of vertices involving ghost fields has been adpated to the conventions of GOSAM.[5] In this example no implicit or explicit power of $\alpha_s$ has been assigned to $O_{\phi G}$, nor any loop-suppression. As a consequence the effective $hgg$-vertex enters the process through one-loop diagrams considered to be of the same order as the ordinary SM virtual QCD corrections.

## 4.4 $H \rightarrow b\bar{b}$ with four-fermion operators

The example Hbb_4F calculates part of the decay of a Higgs boson into a massive $b$-quark pair including the four-fermion operators $O_{qb}^{(1)}$, $O_{qb}^{(8)}$, $O_{qtqb}^{(1)}$ and $O_{qtqb}^{(8)}$. The considered one-loop diagrams are restricted to contain at least one four-fermion vertex, demonstrating the automatic calculation of diagram signs when four-fermion operators are present. The interference of these loop diagrams with the Standard Model Born amplitude is compared to the analytical expression from [71] with a regularisation scheme conversion factor from [72].

## 4.5 $gg \rightarrow Hg$ in SMEFT

The example ggHg_SMEFT describes Higgs+jet production at LO QCD in the loop-induced gluon-gluon channel, including SMEFT effects. It shows how the different SMEFT truncation

---

[5]See section 3.5.1 in the reference manual.

options for loop-induced processes can be used (see section 3.3.4). As for the `Hbb_SMEFT` example the UFO model has been generated using SMEFTFR [23–25], with similar modifications and simplifications. It contains the operators $O_{t\phi}$ and $O_{\phi G}$. For this specific example the latter is assigned both an implicit power of $\alpha_s^2$ and a loop-suppression using the `QL` order feature. In this way the tree-level diagrams generated by this operator contribute at the same perturbative order as the one-loop Born diagrams of the SM.

## 4.6   $gg \to Hg$ in the SM

The example `ggHg_rescue` computes the loop-induced Higgs+jet production process at LO QCD in the SM. The example phase-space point and the requested rescue system thresholds `PSP_chk_li1` – `PSP_chk_li5` are chosen to trigger the quadruple rescue system, which is enabled on the run card of this example by the setting `extensions=quadruple`. This example checks the use of the new rescue system on the user's system and demonstrates the setting of rescue system thresholds. The test comparison values were obtained using a dedicated run of GOSAM in quadruple precision.

## 4.7   Dimensional reduction versus 't Hooft–Veltman scheme

By default GOSAM uses the dimensional reduction scheme (DRED) for the calculation of the amplitude. In order to convert the (renormalised) result into the 't Hooft–Veltman (tHV) scheme the runtime boolean parameter `convert_to_thv` is available. The conversion is carried out using the formulae derived in [73]. It is also possible to carry out the whole calculation in the tHV scheme right from the start, instead. In order to show the two different approaches the example `udeneg_dred_vs_thv` has been added. It calculates the process $u\bar{d} \to \nu_e e^+ g$ through $W$-exchange at NLO in QCD, once in DRED and once in tHV. Note that due to the axial coupling a finite renormalisation of $\gamma_5$ has to be taken into account when calculating in tHV. GOSAM does this automatically.

## 5   Conclusions

We have presented a major upgrade of GOSAM, a program package for the automated generation and evaluation of one-loop amplitudes. The new version comes with a much lighter installation procedure and significant improvements in speed for both, compilation and runtime. The rescue system for numerically unstable points has also been overhauled.

Furthermore, an important new feature is its capability to perform calculations within HEFT or SMEFT, where the user has full control over truncation orders and potential loop suppression of operators. The corresponding model files can be imported via the UFO interface standard [27]. Particular attention has been given to an unambiguous treatment of the diagram sign in the presence of vertices involving more than two fermions.

The capacity of the code to provide one-loop amplitudes within theories beyond the Standard Model is not limited to the EFT case, any model can in principle be imported via the UFO interface. However, a fully automated renormalisation procedure is only provided for SM QCD corrections. Nonetheless, counterterms for the EFT Wilson coefficients can be provided by means of UFO files containing the counterterms.

GOSAM has a flexible interface to Monte Carlo programs providing the real radiation and the infrared subtraction terms, following the BLHA2 [40] standard.

The code comes with a large set of examples demonstrating the various features, and with a reference manual containing detailed documentation.

For future versions of GoSam-3, a higher level of automation of the renormalisation beyond QCD is foreseen, as well as functionalities to include renormalisation group running of the Wilson coefficients.

## Acknowledgements

We would like to thank all former GoSam authors, in particular Stephan Jahn, Pierpaolo Mastrolia, Giovanni Ossola, Tiziano Peraro, Thomas Reiter, Johannes Schlenk, Ludovic Scyboz and Francesco Tramontano. We are also grateful to Jürgen Reuter and Andrii Verbytskyi for useful comments on previous versions of GoSam.

**Funding information**    BC, GH, MH, JL and VM acknowledge support by the Deutsche Forschungsgemeinschaft (DFG, German Research Foundation) under grant 396021762–TRR 257. JB is supported in parts by the Federal Ministry of Technology and Space (BMFTR) under grant number 05H24VKB. SJ is supported by a Royal Society University Research Fellowship (Grant URF/R1/201268) and by the UK Science and Technology Facilities Council (Contract ST/X000745/1).

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
