# Peer review of "One-Loop Calculations in Effective Field Theories with GoSam-3.0"

_SciPost Physics Codebases_

## Round 1 · Referee Report · Anonymous (Referee 1) · 2025-9-19

Strengths
- Importance of the topic
- Usefulness of the software
- Very clear treatment of heigher order in EFT and loop corrections
- Generally clear and well structured
Weaknesses
- Treatment of the Electroweak corrections
- Minor writing errors
Report
Dear Editor,
The paper presents an updated version of the one-loop generator GoSam, an useful tool for automatic and semi-automatic calculations of loop corrections. Compared with MadGraph, it offers distinctive features, and overall represents a meaningful addition to the computational resources available for theoretical high-energy physics. The new version of this program improves runtimes and offers upgraded functionalities, such as the possibility to filter diagram classes and to control truncation scheme. These are particularly useful tools for the calculation of 1-loop amplitudes in general BSM theories, and, specifically, in the SMEFT.
I believe that the structure and content if the paper definitely meet this journal’s criteria for pubblication. Indeed, it is generally clear and well structured, nicely explaining most of the capabilities of the new version, specifically those related to the calculation of QCD corrections. However, the program’s capabilities in calculating electroweak corrections is significantly less clear. While the authors state in the introduction that these functionalities have been extended, the paper provides no further details on the specific features or their implementation.
Concretely, I believe that the authors should clarify: 1. Whether full electroweak one-loop corrections are supported for general models, and if not, what the limitations are. 2. Whether the pole test in the rescue system is applied also to QED poles, since only QCD correction are named, and if not, the rationale. 3. The treatment of gamma_5, which is particularly delicate for electroweak corrections in the SMEFT, due to the structure of the EFT vertices. I would appreciate a more detailed explanation of the scheme used and its implementation.
Finally, I noticed two minor writing mistakes: 1. In the introduction, there is a sentence without a verb: “Additional functionalities, for example enhanced support for multi-leg processes or flexibility concerning the interface to NLO-capable Monte Carlo event generators such as […].” 2. On page 4, in the sentence “If no prefix is given meson will try to install the libraries in /usr/local,” the word meson was written with a different font than the rest of the code-style references.
Finally, as the paper meets the journal’s criteria, I believe that once the authors clarify the status of the electroweak corrections, the manuscript will be suitable for publication in SciPost.
Requested changes
- Clarify the electroweak capabilities of the software and specify the limitations of its use in this context.
Recommendation
Ask for major revision

Author: Gudrun Heinrich on 2025-11-11 [id 6011]
(in reply to Report 1 on 2025-09-19)We thank the referee for the careful reading of the manuscript and the constructive comments. We addressed the comments in the following way:
The referee writes:
We have added the following sentence in the introduction: ``For electroweak corrections, the loop amplitudes can be generated, however the renormalisation is not automated in GoSam-3.0, this is work in progress.''
While the QED IR operator is present in the code, the pole test is disabled for QED poles. This is in line with the fact that the pole test compares IR poles, which can only match after the amplitude has been renormalised. The rescue system for electroweak corrections will be added in version-3.1 together with the electroweak renormalisation. We have added a comment about this at the end of section 3.2.
We added the following paragraph at the end of section 3.3.1:
To ensure a consistent treatment of axial and pseudo-scalar currents, the calculation should be performed in DRED. GoSam will then automatically split Dirac-matrices into their four- and epsilon-dimensional parts as described in Ref.~[11]. The option \texttt{convert_to_thv} allows one to translate the results into the 't~Hooft--Veltman~(tHV) scheme. A direct calculation in the tHV scheme is currently only supported for calculations in the SM using the model files provided by GoSam, where the additional finite renormalisation related to gamma_5 is included automatically~[11].
We also modified the text below equation (37) and in section 4.7 to clarify the possible scheme choices.
We have added "are available in GoSam''.
We have now written it coherently as {\sc Meson} whenever the program rather than the command is meant.

---

## Round 2 · Author Response

We thank the referee for the careful reading of the manuscript and the constructive comments. We addressed the comments in the following way:
The referee writes:
The authors should clarify whether full electroweak one-loop corrections are supported for general models, and if not, what the limitations are.
Our answer We have added the following sentence in the introduction: ``For electroweak corrections, the loop amplitudes can be generated, however the renormalisation is not automated in GoSam-3.0, this is work in progress.''
The referee writes:
Whether the pole test in the rescue system is applied also to QED poles, since only QCD correction are named, and if not, the rationale.
Our answer While the QED IR operator is present in the code, the pole test is disabled for QED poles. This is in line with the fact that the pole test compares IR poles, which can only match after the amplitude has been renormalised. The rescue system for electroweak corrections will be added in version-3.1 together with the electroweak renormalisation. We have added a comment about this at the end of section 3.2.
The referee writes:
The treatment of gamma_5, which is particularly delicate for electroweak corrections in the SMEFT, due to the structure of the EFT vertices. I would appreciate a more detailed explanation of the scheme used and its implementation. Our answer We added the following paragraph at the end of section 3.3.1:
``To ensure a consistent treatment of axial and pseudo-scalar currents, the calculation should be performed in DRED. GoSam will then automatically split Dirac-matrices into their four- and epsilon-dimensional parts as described in Ref.~[11]. The option \texttt{convert\_to\_thv} allows one to translate the results into the 't~Hooft--Veltman~(tHV) scheme. A direct calculation in the tHV scheme is currently only supported for calculations in the SM using the model files provided by \gosam, where the additional finite renormalisation related to gamma_5 is included automatically~[11].''
We also modified the text below equation (37) and in section 4.7 to clarify the possible scheme choices.
The referee writes:
Finally, I noticed two minor writing mistakes: \ 1. In the introduction, there is a sentence without a verb: ``Additional functionalities, for example enhanced support for multi-leg processes or flexibility concerning the interface to NLO-capable Monte Carlo event generators such as […].''
Our answer We have added ``are available in GoSam''.
The referee writes:
- On page 4, in the sentence ``If no prefix is given meson will try to install the libraries in /usr/local,'' the word meson was written with a different font than the rest of the code-style references.
Our answer We have now written it coherently as {\sc Meson} whenever the program rather than the command is meant.

---

## Round 2 · List of Changes

Editorial decision:
For Journal SciPost Physics Codebases: Publish
(status: Editorial decision fixed and (if required) accepted by authors)

---

## Editorial Decision

editorial_decision: